# Facile Attachment of Halides and Pseudohalides to Dodecaborate(2-) via Pd-catalyzed Cross-Coupling

**DOI:** 10.3390/molecules28073245

**Published:** 2023-04-05

**Authors:** Mahmoud K. Al-Joumhawy, Jui-Chi Chang, Fariba Sabzi, Detlef Gabel

**Affiliations:** School of Science, Constructor University, 28759 Bremen, Germany

**Keywords:** dodecaborate(2-), dodecahydrido-closo-dodecaborate(2-), Pd catalysis, halogen exchange, click reaction

## Abstract

Cross-coupling reactions with [B_12_H_11_I]^2−^ as one partner have been used successfully for Kumada and Buchwald Hartwig couplings with Pd catalysis. Here, we found that the iodide could be substituted easily, and unexpectedly, with other halides such as Br and Cl, and with pseudohalides such as cyanide, azide, and isocyanate. We found that for Cl, Br, N_3_, and NCO, tetrabutylammonium salts—or sodium salts—were successful halide sources, whereas for cyanide, CuCN was the only halide source that allowed a successful exchange. The azide could be reacted further in a click reaction with triazoles. While no substitution with fluoride occurred, tetrabutylammonium fluoride in the presence of water led to [B_12_H_11_OH]^2−^. Yields were high to very high, and reaction times were short when using a microwave oven as a heating source.

## 1. Introduction

Dodecaborate clusters are of increasing interest because of their special supramolecular properties: they are water soluble due to their inherent charge, and yet they can interact strongly with hydrophobic surfaces. This can be used in, e.g., their binding to the interior of cyclodextrins [1,2,3,4], or to the outside surface of cucurbiturils [5], and it also manifests itself in large retention factors in chromatography on hydrophilic matrices in water [6]. They can form ionic liquids even with Li^+^ as the cation [7]. At the basis of these phenomena is the weak interaction of the cluster with water as a solvent [8]. While a proper choice of counterion allows good to excellent solubility in water, the solvent interacts only weakly and can therefore be removed easily. This phenomenon makes dodecaborate compounds the first example of superchaotropic agents [4,9].

Such properties are not found in organic compounds. Making full use of these properties requires that the boron cluster is attached to other (organic, organometallic, or inorganic) units. This requires that suitable functionalization methods allow the dodecaborate to react with other functional groups in other building blocks, so that a linkage between them can combine properties from both components. In the past, known types of reactions have been rather limited, and they were achieved mostly by attaching organic moieties to heteroatoms such as O [10,11], N [12,13,14], and S [2,15] (for an overview, see Figure 1).

Recently, we have found that Pd-catalyzed cross-coupling with anilines and amides of monoiodo-dodecaborate ([B_12_H_11_I]^2−^) leads to the formation of B–N-linkers between the dodecaborate and the organic moiety [16]. Previously, [B_12_H_11_I]^2−^ was reacted under Kumada coupling conditions with Grignard reagents [17,18]. The procedure to introduce an iodine atom into the [B_12_H_12_]^2−^ cluster could be improved considerably by using not molecular iodine, but N-iodosuccinimide, resulting in very clean stoichiometric iodinations [19]. The corresponding N-chloro- and N-bromosuccinimides result in the corresponding chloro- and bromododecaborates, again in clean reactions with well-controllable stoichiometry. The [B_12_H_11_NH_3_]^−^ was obtained before by reaction with hydroxylamine-*O*-sulfonic acid; using a Buchwald–Hartwig cross-coupling, we could obtain [B_12_H_11_NH_3_]^−^ from [B_12_H_11_I]^2−^ with urea as source of nitrogen, followed by acid-catalyzed hydrolysis.

While the selective introduction of a single halogen atom is possible, the formation of [B_12_H_11_OH]^2−^, described before [20], uses strongly acidic conditions; it therefore usually leads also to di-substituted products as byproducts [20]. Other dodecaborate substituents such as cyanide, azide, cyanate, and thiocyanate have not been described. The isocyanate derivative has been obtained before through a Curtius rearrangement of a carbonyl azide [21].

In the course of our continued exploration of the further reactivity of [B_12_H_11_I]^2−^, especially with respect to other Pd-catalyzed reactions, reactions with apparently innocent salts, such as NaBr and tetrabutylammonium bromide, occurred. We were very surprised to find that an exchange of the iodide with other nucleophiles was possible, and we could obtain compounds which had not been obtained before. Several of these compounds can be reacted further, while others might be unwanted side products in further reactions.

## 2. Results

Our investigation started with an observation when we tried to run Pd-catalyzed cross-coupling reactions in a heterogeneous system of water and immiscible solvents. In order to increase the solubility of water-soluble anions, we wanted to resort to phase transfer catalysts such as Bu_4_NBr to transfer the anion into the organic phase. To our surprise, even in the absence of any further reagent (except KOH, which would serve as a base for other reactions) we found conversion of [B_12_H_11_I]^2−^ to the corresponding [B_12_H_11_Br]^2−^. Halogen exchange is known to occur in aryl halides [22], but we had not observed this reaction when further, probably better, groups with good ability to interact with the Pd cation were present. We therefore checked further salts, first with the tetrabutylammonium cation, and then with Na^+^ as the counterion.

Successful exchange reactions could be performed using Pd_2_(dba)_3_ as the Pd source and Davephos as the ligand in DMSO, as previously found for Buchwald–Hartwig type cross-couplings (Ref. [16]). For the reaction with the halides and pseudohalides Br^−^, Cl^−^, and N_3_^−^, the choice of counterion (Bu_4_N^+^ or Na^+^) did not matter, and the corresponding derivatives were obtained in good yield. Depending on the workup, there might be a partial or complete exchange of the cation when starting with Bu_4_N^+^ salts of the cluster and using a sodium halide as a halide source.

Reactions were considerably faster, at identical temperatures, when conducted with microwave heating. This is in agreement with our previous experience with substitution reactions on dodecaborates [16,23]. We attribute this to the inherent dipole of the cluster and the ionic conduction, both of which are known to increase the energy transfer of microwave radiation into the solution. 

We then checked whether KOH as base (required for Buchwald–Hartwig or Suzuki reactions) was necessary by leaving it out, and the reaction proceeded equally well without KOH. Figure 2 shows the conditions and the scope.

We were able to obtain an isocyanate using NaNCO as a cyanate donor. This compound has been obtained before by the reaction of the CO derivative of dodecaborate [24] with sodium azide through a Curtius rearrangement [21]. On the basis of the ^11^B NMR spectrum, we identified the N atom as the B-bonded atom, which resonates at −10 ppm in agreement with the product described in the literature [21]. For oxygen compounds bound to B, boron-11 resonances at positive ppm values (relative to BF_3_ etherate) are observed [20,25], while the compound isolated here, as all other B-N cluster bonds, has a resonance at negative ppm values for the heteroatom-substituted B atom.

The analogous reaction with thiocyanate did not provide any product. This might, in part, be attributed to strong interactions between the sulfur of the thiocyanate and the Pd atom (observed for similar structures not containing boron [26]), preventing any further reactions. The thiocyanate had been obtained before by the reaction of the dirhodanide pseudohalogen with [B_12_H_12_]^2−^ [27], in a reaction similar to other halogenation reactions.

When trying to exchange the iodide of [B_12_H_11_I]^2−^ with F^−^, using anhydrous tetrabutylammonium fluoride, or NaF, we did not succeed in obtaining any substitution with fluoride. When using the trihydrate of the Bu_4_NF and KOH as a hydroxide source, a very clean reaction to [B_12_H_11_OH]^2−^ occurred. Given the ease of preparing [B_12_H_11_I]^2−^ in its pure monosubstituted form with N-iodosuccinimide and its smooth conversion to [B_12_H_11_OH]^2−^ described here, this route might be preferable to the routes described in the past, which used aqueous acid.

The hydroxy derivative was a side product when using the common hydroxides (sodium, potassium, tetrabutylammonium—the latter either in water or in THF). With these reagents, only a little hydroxydodecaborate (20% at most) was formed. This also manifests itself in the observation that KOH is a good base for cross-coupling reactions, both here and in the Buchwald–Hartwig reaction, without the formation of hydroxydodecaborate [16]. This is in contrast with the reaction of aryl halides with KOH in water, which, under Pd catalysis, leads to phenols in excellent yields [28].

For CN^−^ as anion, neither tetrabutylammonium nor sodium as countercations gave the desired product. In addition, KCN, Zn(CN)_2_, and K_4_[Fe(CN)_6_] did not provide any cyano products. Only its copper(I) salt could be used successfully (Figure 2).

Previously, the complete replacement of the iodine atoms on [B_12_I_12_]^2−^ with CN^−^, using Pd salts and prolonged high-temperature heating with microwave irradiation, has been reported by Kamin et al. [29]. A bulky Pd ligand, *t*BuPrettPhos, more space demanding than the Davephos used here, was needed for achieving reasonable yields.

Further reactions of the new compounds obtained here (see Figure 3) are also of great interest. 

The azide obtained here for the first time invites one to perform click chemistry. While this has been described for carboranyl azides [30], such reactions have not yet been described for the dodecaborate cluster. As a prototype reaction, we used acetylene dimethylcarboxylate. The desired product was obtained in excellent yield after only short reaction times and a simple workup.

The isocyanate (described previously and used for reactions by others [21]) is known to react to the corresponding urea derivative with amines. We tested whether we could also cause a reaction with alcohols under basic conditions to the corresponding carbamates. To our surprise, alkoxides could not react to form the desired product. We attribute this to the large electron density donated by the cluster to the carbonyl carbon. We have seen previously that the cluster considerably increases the pK_a_ values of the –SH and the –NH_2_ groups attached to it [15,31]. 

## 3. Discussion

The exchange reactions described here work only with [B_12_H_11_I]^2−^ as the starting material. The bromide and the chloride did not react; we had already observed this for cross-coupling reactions with amides and anilines [16]. This is in contrast to carboranes, where the bromide can undergo reactions to the isocyanate and azide, and to the amides [30]. 

We speculate that the exchange with the halides and pseudohalides happens after the oxidative insertion of the Pd into the B–I bond, similar to the mechanisms proposed for the aryl halide exchange [22]. The iodide that had been bound originally to the cluster can exchange with the incoming halide or pseudohalide, and the resulting complex undergoes reductive elimination to the final product. While one would expect this to work in both directions, we found before that in contrast to [B_12_H_11_I]^2−^, the chloro- or bromoderivatives were non-reactive in cross-coupling with Pd under Buchwald–Hartwig conditions [16]. Thus, even if the formation of the bromo- or chloro-dodecaborate might not be an energetically favored reaction, due to the low reactivity of the product under cross-coupling conditions, the oxidative insertion of Pd into the B–Br or B–Cl bond will not be possible, and the product will therefore accumulate. The fluoride might be too hard a nucleophile to bind well to the Pd (which is known to be a soft electrophile) and to replace the iodide in the complex.

Our results might also be of importance when salts are added as presumably non-reacting additives to cross-coupling reactions with [B_12_H_11_I]^2−^, as the anions of the salts might react preferentially.

The surprising formation of [B_12_H_11_OH]^2−^ offers a new route to this compound, which can act as a nucleophile and be alkylated or acylated [11].

The preparation of the isocyanate has been achieved before through a multistep reaction from the carbonyl-substituted cluster [21]. The method proposed here is an alternative, which might result in an easier preparation of this compound.

While the bromo- and chloro-dodecaborates will be largely chemically inert, the pseudohalogens azide, isocyanate, and cyanide can react further. The reaction of the isocyanate with amines is known [21]. Even more interesting for further reactions is the click reaction of the azide with alkynes. As the click reaction occurs under mild conditions, and there are plenty of potential alkyne reaction partners, their conjugation with the dodecaborate and its unique properties might open up new opportunities in cellular transport of attached biological effectors [32], in covalent stabilization of non-covalent networks [33], and in further applications.

## 4. Materials and Methods

Dodecahydrido-*closo*-dodecaborate was purchased as Cs^+^ salt from BASF. It was converted to tetrabutylammonium salt by precipitation from an aqueous solution with Bu_4_NCl dissolved in water. All other chemicals and solvents were from Sigma-Aldrich, St. Louis, MO, USA, or Carl Roth, Karlsruhe, Germany, and were used as received. 

^11^B NMR spectra were recorded on a JEOL 400 MHz spectrometer at 25 °C. Chemical shifts were referenced relative to external BF_3_ etherate. MestReNova V10.0.2-15465 S3 software (Mestrelab Research, Santiago de Compostela, Spain) was used to visualize the spectra. Coupling constants (*J*) are reported in Hertz (Hz). NMR spectra of the new compounds described here are provided in the Appendix A).

Mass spectra were recorded on a Waters QTOF Premier spectrometer in negative mode, using acetonitrile as solvent for electrospray. The *m*/*z* values reported below are those of the most intense peaks. MS spectra of the new compounds described here are provided in the Appendix A).

Reagents and solvents were commercially available and used without further purification. The [B_12_H_11_I]^2−^ (Bu_4_N)_2_ was prepared according to the literature procedure, with some modifications. Tetrabutylammonium bromide, tetrabutylammonium chloride, tetrabutylammonium fluoride trihydrate, copper cyanide, sodium bromide, sodium chloride, tetrabutylammonium azide, sodium azide, Davephos, and Pd_2_(dba)_3_ were purchased from Sigma-Aldrich and were used as received. The DMSO, dichloromethane, acetonitrile, and silica gel (Grade 60, 230–400 Mesh) were from Carl Roth. Celite (545 filter aid, not acid washed, powder) was from Fisher. All cross-coupling reactions were performed in an oven-dried 10 mL round-bottom flask. The thin-layer chromatography (TLC) AluSil plates were from MachereyNagel, Düren, Germany. The TLC samples for borane-containing compounds were stained with 1 wt.% PdCl_2_ in 6 M HCl and were developed using a heat gun. An open-vessel microwave oven (CEM Discover, Model 908860, or HNZXIB, Model MCR-3) was used.

### 4.1. General Procedure

A dry 10 mL round-bottom flask fitted with condenser was charged with 1.0 equivalents of: (Bu_4_N)_2_B_12_H_11_I, Pd_2_(dba)_3_ (5 mol%); Davephos (10 mol%); 3–5 equivalents of tetrabutylammonium bromide or NaBr for synthesis of (Bu_4_N)_2_B_12_H_11_Br; Tetrabutylammonium chloride or NaCl for synthesis of (Bu_4_N)_2_B_12_H_11_Cl; Tetrabutylammonium azide or NaN_3_ for synthesis of (Bu_4_N)_2_B_12_H_11_N_3_; Tetrabutylammonium fluoride·3H_2_O and 5.0 equivalents of KOH were used for the synthesis of (Bu_4_N)_2_B_12_H_11_OH; and copper(I)cyanide was used for the synthesis of (Bu_4_N)_2_B_12_H_11_CN. Subsequently, 2.0 mL of anhydrous DMSO was added. The reaction flask was filled with N_2_ and connected to a condenser. The round-bottom flask was placed in a CEM microwave oven and heated to 150 °C, at a maximum power of 300 W, for 15 min with stirring (high), or heated in an oil bath at 150 °C for 3–5 h until the starting (Bu_4_N)_2_B_12_H_11_I was completely consumed, as judged by ^11^B NMR and TLC. The mixture was cooled to room temperature and then filtered through a funnel filled with cotton, celite, and filter paper. The resulting solution was concentrated under reduced pressure and the crude product was subjected to silica gel chromatography using a gradient of acetonitrile (0–25%) in DCM.

The spectroscopic data of the compounds known from the literature (Br, Cl, OH) agree with the literature data.

**(Bu_4_N)_2_B_12_H_11_Br:** ^11^B NMR (129 MHz, DMSO-*d_6_*, δ): −19.30 (d, *J* = 128 Hz, 1B, B-H), −16.27 (d, *J* = 134 Hz, 5B, B-H), −14.71 (d, *J* = 136 Hz, 5B, B-H), −8.91 (s, 1B, B-Br). HRMS (ESI/TOF) *m*/*z* for B_12_H_11_Br [M]^2−^: 110.55 (found: 110.57).

**(Bu_4_N)_2_B_12_H_11_Cl:** ^11^B NMR (129 MHz, DMSO-*d_6_*, δ): −20.8 (d, *J* = 129 Hz, 1B, B-H), −17.2 (d, *J* = 129 Hz, 5B, B-H), −15.5 (d, *J* = 131 Hz, 5B, B-H), −3.7 (s, 1B, B-Cl). HRMS (ESI/TOF) *m*/*z* for B_12_H_11_Cl [M]^2−^: 88.08 (found: 88.10).

**(Bu_4_N)_2_B_12_H_11_N_3_:** ^11^B NMR (129 MHz, DMSO-*d_6_*, δ): −19.61 (d, *J* = 129 Hz, 1B, B-H), −16.07 (d, *J* = 129 Hz, 5B, B-H), −16.43 (d, *J* = 131 Hz, 5B, B-H), −2.68 (s, 1B, B-N_3_). HRMS (ESI/TOF) *m*/*z* for B_12_H_11_N_3_ [M]^2−^: 91.60 (found: 91.62).

**(Bu_4_N)_2_B_12_H_11_OH:** ^11^B NMR (129 MHz, DMSO-*d_6_*, δ): −25.81 (d, *J* = 129 Hz, 1B, B-H), −19.71 (d, *J* = 125 Hz, 5B, B-H), −17.08 (d, *J* = 124 Hz, 5B, B-H), 4.23 (s, 1B, B-OH). HRMS (ESI/TOF) *m*/*z* for B_12_H_11_OH [M]^2−^: 79.10 (found: 79.10).

**(Bu_4_N)_2_B_12_H_11_CN:** ^11^B NMR (129 MHz, DMSO-*d_6_*, δ): −22.86 (s, 1B, B-CN) −18.60 (d, *J* = 149 Hz, 1B, B-H), −16.43 (d, *J* = 123 Hz, 10B, B-H). HRMS (ESI/TOF) *m*/*z* for B_12_H_11_N_3_ [M]^2−^: 83.60 (found: 83.61).

**(Bu_4_N)_2_B_12_H_11_NCO:** ^11^B NMR (129 MHz, DMSO-*d_6_,* δ): −21.25 (d, *J* = 123.7 Hz, 1B, B-H), −18.72 (d, *J* = 154.14 Hz, 5B, B-H), −17.40 (d, *J* = 154.14 Hz, 5B, B-H), −9.75 (s, 1B, B-NCO). HRMS (ESI/TOF) *m*/*z* for B_12_H_11_NCO [M]^2−^: 91.59 (found 91.60).

### 4.2. Click Reaction

(Bu_4_N)_2_B_12_H_11_N_3_, (1 mmol) was dissolved in CH_3_CN; subsequently, Cu(I)OAc (0.2 mmol) and sodium ascorbate (0.3 mmol) were added. Then, dimethyl but-2-ynedioate (2.0 mmol) was added to the mixture and the reaction mixture was stirred at 85 °C for 3 h. The reaction progress was monitored by TLC and NMR. After the complete consumption of the starting material, the reaction mixture was allowed to cool to room temperature, and the insoluble materials were removed via filtration. The desired product was collected in 95% yield by a short plug of silica gel using 1:1 DCM:Acetonitrile.

^11^B NMR (129 MHz, Acetonitrile-*d_3_*, δ): −5.4 (s, 1B), −16.4 (d, *J* = 50 Hz, 5B), −17.1 (d, *J* = 48.9 Hz, 5B), −18.4 (d, *J* = 83 Hz, 1B). ^1^H NMR (400 MHz, Acetonitrile-*d_3_*, δ): 0.95–0.98 (t, 24H), 1.31–1.40, 16H), 1.56–1.64 (m, 16H), 3.07, 3.09, 3.11 (m, 16H), 3.80 (s, 3H), 3.84 (s, 3H). HRMS (ESI/TOF) *m*/*z* for B_12_H_17_C_6_O_4_N_3_ [M]^2−^: 162.61 (found: 162.60).

## 5. Conclusions

We have found that [B_12_H_11_I]^2−^ reacts with other halides and pseudohalides under cross-coupling conditions to yield compounds which have not been previously obtained, and to yield other products in a simple manner. Several of these compounds can be reacted further, thus considerably broadening the possibilities of linking the dodecaborate cluster to other fragments of interest for biology, drug development, and material sciences.

## Data Availability

All experimental procedures are described in the paper. The NMR and MS spectra are deposited in the Appendix A.

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
