# Peer review of "Facile Attachment of Halides and Pseudohalides to Dodecaborate(2-) via Pd-catalyzed Cross-Coupling"

_molecules, 2023, doi:10.3390/molecules28073245_

Round 1

Reviewer 1 Report

1. In the abstract section, I suggest the author could add more information for the reader.

2. Introduction: "…In the past, known types 34 of reactions have been rather limited, and they were achieved mostly by attaching organic 35 moieties through heteroatoms such as.." Some refs could be updated, such as Chem. Commun., 2022, 58, 6653–6656; Org. Chem. Front., 2020,7, 3515-3520; J. Org. Chem. 2019, 84, 14627−14635 and Org. Chem. Front., 2021, 8, 4554–4559.

3. In the introduction section, please highlight your design idea.

4. I suggest the authors have to list a Table for comparing the previous examples and presented in this work on the Pd-Catalyzed Cross-Coupling.

5. I suggest the authors could do a DFT to support the Cross-Coupling.

6. In Scheme 1, why does the BuNN3/NaN3 have low yield? Please explain.

7. In the conclusion part, the authors also have to illustrate the challenge in this topic.

Reviewer 2 Report

1. The quality of the figures should be improved.

2 I would appreciate a small section concerning with the rational designing of Pd-Catalyzed Cross-Coupling.

3. There are some typos in the paper. Please correct in the revised version.

4. Some related refs on such topic could be cited, such asMolecules, 2019, 24, 1760 and J. Org. Chem. 2019, 84, 14627−14635;Org. Chem. Front., 2021, 8, 4554–4559

5. The authors should compare/construe the presented literature information, emphasize important achievements and provide with and outlook regarding likely future trends. 

Round 2

Reviewer 1 Report

accept